# The Potential Reversible Transition between Stem Cells and Transient-Amplifying Cells: The Limbal Epithelial Stem Cell Perspective

**DOI:** 10.3390/cells13090748

**Published:** 2024-04-25

**Authors:** Sudhir Verma, Xiao Lin, Vivien J. Coulson-Thomas

**Affiliations:** 1College of Optometry, University of Houston, 4901 Calhoun Road, Houston, TX 77204, USA; xiao.lin@bcm.edu; 2Deen Dayal Upadhyaya College, University of Delhi, Delhi 110078, India

**Keywords:** dedifferentiation, transit-amplifying cells (TACs), limbal epithelial stem cells (LESCs), stem cells, progenitors, cornea, centrifugal movement

## Abstract

Stem cells (SCs) undergo asymmetric division, producing transit-amplifying cells (TACs) with increased proliferative potential that move into tissues and ultimately differentiate into a specialized cell type. Thus, TACs represent an intermediary state between stem cells and differentiated cells. In the cornea, a population of stem cells resides in the limbal region, named the limbal epithelial stem cells (LESCs). As LESCs proliferate, they generate TACs that move centripetally into the cornea and differentiate into corneal epithelial cells. Upon limbal injury, research suggests a population of progenitor-like cells that exists within the cornea can move centrifugally into the limbus, where they dedifferentiate into LESCs. Herein, we summarize recent advances made in understanding the mechanism that governs the differentiation of LESCs into TACs, and thereafter, into corneal epithelial cells. We also outline the evidence in support of the existence of progenitor-like cells in the cornea and whether TACs could represent a population of cells with progenitor-like capabilities within the cornea. Furthermore, to gain further insights into the dynamics of TACs in the cornea, we outline the most recent findings in other organ systems that support the hypothesis that TACs can dedifferentiate into SCs.

## 1. Introduction

Stem cells were first identified in the bone marrow, which are now known as hematopoietic stem cells [1,2,3,4,5,6,7,8]. Since then, a number of stem cells have been identified in various tissues throughout the body, where they are able to maintain homeostasis and regenerate tissues after damage [9,10,11,12]. A vital property of stem cells is the capability to undergo asymmetric division, producing a stem cell that maintains the stem cell pool and another cell that moves out of the stem cell niche and differentiates into a specialized cell type. A critical step in this process is the intermediary state between the stem cell and the differentiated cell, which is the transit-amplifying cell (TAC). Thus, TACs represent the transition state between an undifferentiated cell and a cell that is committed to a certain lineage [13]. Stem cells are deemed quiescent cells; however, TACs are characterized by rapid proliferation to generate new cells that are required to maintain/regenerate the tissue [13,14]. Over the years, substantial focus has been dedicated to identifying stem cells within tissues and characterizing the stem cell niche; however, significantly less focus has been dedicated to the TACs. Understanding the mechanisms that govern the transition of TACs from the stem cell state to the differentiated state is crucial for understanding how stem cells maintain tissue homeostasis and regenerate tissues after injury, and it is crucial for developing stem cell-based therapies. Furthermore, recent studies have suggested that TACs could represent a reversible state between stem cells and differentiated cells; thus, understanding the mechanisms that govern the potential dedifferentiation of TACs into stem cells would significantly advance the field of regenerative medicine.

The cornea is the outer most part of the eye, which serves as a protective barrier against continuous environmental insults; however, despite this, it must remain transparent to allow the passage of light into the eye for vision. A vital property of the cornea that allows it to maintain transparency throughout life is the existence of stem cell populations within the limbal region, including the limbal epithelial stem cells (LESCs). LESCs reside in the transition zone between the transparent cornea and the conjunctival epithelium [15], in a specific niche called the limbal stem cell niche (LSCN). The LSCN is comprised of cellular elements, such as mesenchymal cells, immune cells, melanocytes, nerve and vascular cells; extracellular matrix (ECM) components, including hyaluronan (HA); and signaling molecules [16]. The importance of LESCs for maintaining a healthy cornea is evidenced by the consequence of the loss of LESCs, which leads to limbal stem cell deficiency (LSCD), a condition characterized by conjunctivalization of the cornea, chronic epithelial erosions, chronic inflammation, neovascularization, and severe pain [17]. Recently, various studies have reported that progenitor cells exist in the central cornea that could also participate in corneal epithelial regeneration [18,19,20]. Given that TACs represent a transition state between stem cells and differentiated cells, and as they exist in the cornea, they could represent the progenitor-like cell population that has been suggested to exist within the cornea. Furthermore, TACs have high proliferative activities and extensive cell population expansion capabilities [21]. Although much is known about LESCs and their role in corneal homeostasis and repair, relatively little is known about the TACs. Herein, this review will focus on recent advances in our understanding of TACs, particularly in the cornea. Specifically, we have summarized the evidence in support of the existence of progenitor-like cells in the cornea and discussed the possibility of these progenitor-like cells being TACs, and whether these TACs with progenitor-like properties are able to dedifferentiate into SCs under certain circumstances. To gain further insights into the dynamics of TACs in the cornea, we also discussed the mechanisms by which the transition between SCs and TACs is capable of maintaining homeostasis and regeneration in other tissues. 

## 2. Maintenance of Corneal Epithelia during Homeostasis and Following Wounding: The Role of LESCs and TACs

The corneal epithelium is formed of a stratified epithelium that is continuously renewed throughout life, with an estimated turnover rate of one week for humans [22]. Up until the discovery of adult stem cells, the corneal epithelium was believed to be a tissue capable of self-renewal [23]. In the 1960s, Hanna and O’Brien showed that basal corneal epithelial cells divide and move vertically through the epithelial layers and eventually slough off within 3.5–7 days, based on 3H-TdR labeling in mice and rats [24], suggesting the proliferation of basal epithelial cells maintains the corneal epithelium. By 1983, a model of cell movement, popularly known as the ‘XYZ hypothesis’, was proposed by Thoft and Friend [25]. The ‘XYZ hypothesis’ states that if the proliferation of basal epithelial cells is X, the contribution to the cell mass of the centripetal movement of peripheral cells is Y, and the epithelial cell loss from the surface is Z, then corneal epithelial maintenance can be defined as: X + Y = Z, thereby proposing that to maintain homeostasis, the epithelial cell loss must be balanced by cells moving in from the limbal region and by the proliferation of basal epithelial cells. Originally, this model did not directly consider the contribution of LESCs to corneal epithelial maintenance, but it provided the framework for developing the LESC hypothesis proposed in 1986 [26]. We now know that the Y component of the original XYZ model represents the TACs that move into the cornea from the limbus. In 2012, Mort et al. suggested the XYZ hypothesis be redefined as: X_TAC_ + Y_SC_ = Z_L_, where X_TAC_ is the proliferation of the basal corneal epithelial (believed to be the TACs), Y_SC_ is the new TACs that move into the cornea, and Z_L_ is the epithelial cell loss from the surface [23]. Upon injury, there is an increased Z, which requires an increase in basal cell proliferation (X) and/or increased centripetal cell movement (Y) in order to resurface the corneal epithelium [25]. 

LESCs undergo asymmetric division, wherein one daughter cell remains as an LESC and is retained within the limbal stem cell pool, while the other differentiates into TACs (with high but limited proliferative activity), detaches from the LSCN, and moves centripetally into the cornea [15] (Figure 1A). As TACs move into the cornea, they progressively lose their proliferative potential, and ultimately, differentiate into differentiated corneal epithelial cells [27]. Based on their location and proliferative capacity, TACs are divided into early or young TACs and late or mature TACs. Early/young TACs exist in the limbal region and peripheral cornea and have considerable proliferative capacity, whereas late/mature TACs exist in the central cornea and undergo limited (1–2) rounds of division. These findings also indicate that under homeostasis, a TAC does not use all its replicative capacity prior to becoming a post-mitotic differentiated cell. However, upon injury/wounding, TACs undergo increased amplification divisions to repair the defect, to the extent that they deplete their proliferative potential [21]. Lehrer et al. also showed that during homeostasis, TACs have a relatively long cell cycle time of about 72 h and replicate at least twice [21]. But in response to a corneal injury, these TACs reduce their cell-cycle time, undergoing additional cell divisions. Interestingly, these cell-cycle changes occur concomitantly with changes in the distribution of cells that express putative LESCs and/or early TAC markers. For example, CCAAT/enhancer binding protein (C/EBPδ), polycomb complex protein-Bmi1 and the N-terminal truncated form of p63α (ΔNp63α)-positive basal limbal epithelial cells during normal homeostasis are considered quiescent LESCs (qLESCs). However, following corneal injury, some of these cells lose C/EBPδ and Bmi1 expression while maintaining ΔNp63α expression, as they proliferate and move into the cornea, suggesting that some qLESCs become active LESCs (aLESCs) and early TACs [28]. On the other hand, in the case of larger wounds, integrin α9-expressing TACs primarily move to repair the wound, causing a depletion of integrin α9-expressing basal limbal epithelial cells [29].

## 3. Evidence of the Existence of Progenitor-like Cells in the Peripheral Cornea

Recent studies have indicated that upon depletion of a stem cell pool, in some tissues, stem cells can be regenerated by the dedifferentiation of differentiated cells into stem-cell-like cells [30,31]. Substantial data have emerged to show that this phenomenon also occurs in the cornea, with studies demonstrating that upon injury to the limbal region that depletes the limbal epithelial stem cell population, corneal epithelial cells can move centrifugally to repopulate the limbal region [32]. Furthermore, the corneal epithelial cells that move into the limbal region, dedifferentiate, and attain an expression profile similar to that of LESCs [32] (Figure 1B). This phenomenon was first suggested by Majo et al. in 2008, identifying that following the depletion of epithelial cells within the limbal rim, central cornea grafts transplanted onto the limbal region exhibit a remarkable ability to repopulate the limbal region and maintain a transparent cornea over extended periods [19]. This controversial study showed that the cornea remains transparent following complete depletion of the limbal stem cell population by cauterization, suggesting that LESCs are not necessary for maintaining the corneal epithelium during homeostasis [19]. Li and colleagues then demonstrated that BrdU label-retaining cells that express putative limbal epithelial stem cell markers exist in both the limbal region and cornea, suggesting progenitor-like cells to exist in the corneal epithelium of mice [33]. Nasser et al. recently demonstrated that following a full limbal debridement wound, epithelial cells within the peripheral cornea move centrifugally to repopulate the limbal region. Furthermore, within the limbal region, these corneal epithelial cells proceed to stratify, and within 10 days following wounding, a subset of the basal cells within the limbal region proceed to express the putative LESC marker K15 using a transient ‘on-off’ GFP reporter for K15 promoter activity and maintain this K15 expression pattern for up to at least 4 months [34]. Importantly, in this study, a debridement wound was used to deplete the limbal epithelial stem cells, thus maintaining the underlying stromal niche intact [34]. However, when the limbal region is depleted via an alkali burn confined to the limbal region, the corneal epithelial cells do not proceed to express putative LESC markers as they move into the limbal compartment, indicating that LSCN factors regulate this dedifferentiation process [34]. Nasser et al. demonstrated that corneal epithelial cells that repopulate the limbal epithelium prevent goblet cells from moving into the cornea, thereby preventing conjunctivalization [34]. Furthermore, if both the limbal epithelium and the corneal epithelium are removed via debridement, conjunctival cells move into the limbal region and resurface the cornea, and the cornea turns opaque and vascularized [35]. The conjunctival cells that move into the limbal region do not proceed to express putative LESC markers, indicating conjunctival cells are not reprogrammed into LESC-like cells after entering the limbal compartment. Instead, when conjunctival cells bypass the limbus in pathological conditions such as LSCD, they result in goblet cell metaplasia, which is correlated with the presence of neovascularization [36]. However, in the absence of neovascularization, the corneal epithelial can repair the injured surface [36]. 

The peripheral cornea epithelium has also been shown to exhibit significant proliferative capability in responding to injuries, forming an epithelial cell population pressure that can drive the resurfacing of a central debridement wound [37]. In fact, studies have found comparable proliferative and migration rates between central (0.06 ± 0.01 mm/h) and peripheral corneal epithelial cells (0.07 ± 0.03 mm/h) as they resurface peripheral wounds, irrespective of the limbal epithelium removal within the initial 12 h post-wounding. This further indicates that corneal epithelial cells have proliferative potential that can drive the resurfacing of a corneal epithelial injury without the involvement of LESCs [20]. In fact, we have previously shown that the cornea can heal small debridement wounds without the involvement LESCs [38]. Goodell and colleagues first demonstrated that a subset of mouse hematopoietic stem cells presented the unique ability to efflux the DNA-binding dye Hoechst 33342, primarily based on their high expression of ABC (ATP-binding cassette) transporters that could actively pump the dye out of the cell [39]. Based on this assay, a subpopulation of cells existed within the hematopoietic stem cells pools that could be cell-sorted based on their low-fluorescence staining pattern, which presented long-term multi-lineage reconstitution abilities, and these cells were described as a side population. This side population has since been identified in other tissues, including mammary glands [40,41], liver [42,43], and lungs [44,45]. A side population of cells that can efflux fluorescent dyes was also identified in the human and rabbit limbal epithelium that express the LESC marker ABCG2; however, a side population was not identified in the cornea [46,47]. In the rat model, a side population was identified in both the limbal region and cornea, which was isolated by fluorescence-activated cell sorting (FACS); however, only the limbal side population presented the expression of putative stem cell markers [32]. 

As mentioned above, the LESC niche was found to be a critical factor for triggering the dedifferentiation of corneal epithelial cells into LESCs [34]. Thus, the capability of the corneal epithelial cells to dedifferentiate into stem-cell-like cells is reliant on factors within the limbal stem cell niche [11]. Therefore, studies are currently underway to identify the key factors within the LSCN that can trigger the dedifferentiation of corneal epithelial cells into LESCs. Studies have demonstrated that exosomes produced by keratocytes [48], miRNA [49], biomechanical properties of ECM [50] and certain ECM components [51,52] can regulate the fate of the corneal epithelial cells. Furthermore, we have identified that HA, a key component of the LSCN, is critical for maintaining murine and human LESCs in the stem cell state, both *in vivo* and *in vitro* [52,53,54]. The basement membrane (BM), a highly specialized acellular layer of extracellular matrix that underlies and interacts with basal epithelial cells, and that regulates their anchoring, migration and differentiation, is another important component of the LSCN [55,56]. Regional differences in the composition of the BM affect cellular activity [57,58,59]. The limbus-specific BM provides a unique microenvironment for the maintenance, self-renewal, activation, and proliferation of LESCs. BM components that are uniformly expressed throughout all the ocular surface epithelia include type IV collagen α5 and α6 chains, collagen types VII, XV, XVII, and XVIII, laminin-111, laminin-332, laminin chains α3, β3, and γ2, fibronectin, matrilin-2 and 4, and perlecan. The limbal and conjunctival epithelium BMs share many similarities, including type IV collagen α1 and α2 chains, laminin α5, β2, and γ1 chains, nidogen-1 and -2, and thrombospondin-4. Components exclusively present in the limbal BM include laminin α1, α2, β1 chains, laminin γ3 chain, agrin, BM40/SPARC, and tenascin-C. Interestingly, these components co-localize with ABCG2/p63/K19-positive and K3/Cx43/desmoglein/integrin-alpha2-negative cell clusters, comprising putative stem and early progenitor cells in the basal epithelium of the limbal palisades. In the corneal–limbal transition zone, XVI collagen, fibulin-2, tenascin-C/R, vitronectin, bamacan, chondroitin sulfate, and versican are present, all of which co-localize with putative late progenitor cells in the basal epithelium [55,58]. Thus, the BM provide a unique microenvironment for LESCs [57,60,61,62]. Taken together, the characterization of the LSCN opens new pharmaceutical avenues using ‘stemness-promoting’ factors to increase or restore the LECSs following injuries [54,63,64,65].

Although the presence of progenitor cells within the murine cornea is currently widely accepted, whether or not these cells exist in the human cornea remains to be established. Studies have shown that in a human corneal organotypic culture model, cells from the corneal epithelium can repopulate the limbal epithelium following a excimer laser-assisted corneal epithelium removal, similar to what was observed with the murine model [20]. However, this study did not verify whether the corneal epithelial cells that move into the limbal region proceed to express LESC markers, which would indicate a potential dedifferentiation process took place [20]. Curiously, some patients who are diagnosed with total LSCD have been shown to maintain a healthy cornea for up to 12 years after diagnosis [18]. This indicates that LESCs are not necessary for maintaining corneal homeostasis, and potentially, that the human cornea could present progenitor-like cells that can contribute toward maintaining a healthy corneal epithelium during homeostasis, as seen in the murine model. Chang et al. (2011) compared the stem cell properties of the cells isolated from central and limbal epithelium of the human cornea and found that both the cells have stem cell properties; however, the central corneal cells lose their stem cell properties with age [66]. Furthermore, cells with colony formation capabilities, indicative of progenitor-like potential, have been identified within the cornea of various mammalian species, including humans [32].

Taken together, the collective evidence strongly suggests the presence of ‘stem-cell-like’ cells within the peripheral and central cornea of the murine cornea. Therefore, based on these studies, the current hypothesis is that a group of ‘stem-cell like’ cells exist within the peripheral or central cornea, characterized by their ability to maintain the corneal epithelium during homeostasis and exhibit increased proliferative potential upon injuries [67]. Furthermore, upon limbal injuries that spare the LSCN, these cells can move centrifugally to repopulate LESCs via a dedifferentiation process. Although the existence of these progenitor-like cells within the cornea is well established, the identity and location of these cells within the cornea remains unknown.

## 4. TACs: The Potential Progenitor-like Cells within the Peripheral Cornea

Although substantial research supports the hypothesis that stem-cell-like or progenitor-like cells exist within the cornea during homeostasis, these cells remain to be identified and isolated from the cornea. Currently, it remains to be established whether there is a separate independent stem cell pool within the cornea, or whether the progenitor-like cells that exist within the cornea are simply TACs derived from LESCs that retain some stem-cell-like properties and are capable of moving back into the limbal region and dedifferentiating into LESCs (Figure 1B).

Recent studies based on label-retaining techniques and scRNAseq indicate that not all LESCs are equal, and instead, two main populations of LESCs have been postulated to exist within the limbal epithelium [68]. A more quiescent LESC population (qLSCs) exists in the ‘outer’ limbus that is believed to serve as a reservoir of LESCs, and a more active population of LESCs (aLSCs) exists in the ‘inner’ limbus that supplies TACs to renew and maintain the cornea during homeostasis [68]. The aLSC population is believed to produce TACs with limited replication potential, capable of undergoing only 3–4 cell divisions, while the qLESCs are triggered to proliferate following corneal injury and are essential for contributing toward corneal regeneration [68]. As TACs are produced, they move centripetally into the cornea, providing a continuous source of young cells. TACs, unlike SCs, are characterized by their rapid cycling, which contributes toward the replenishment of epithelial cell populations as they move toward the central cornea [13,21]. Studies have indicated that different populations of TACs exist in the different zones of the cornea [21,69]. Furthermore, the proliferative capability of TACs has been shown to vary based on their location within the cornea [21,70]. Lehrer and colleagues demonstrated that TACs in the peripheral cornea of mice can replicate at least twice before terminal differentiation, whereas TACs in the central cornea are only able to divide once [21]. In the corneal epithelium, a hierarchy of TACs seems to exist, with division capacity gradually decreasing from the peripheral to central cornea [21]. Also, in both the human and rodent models, p63 signaling was found to be the most intense in the limbal region and to gradually decrease toward the central cornea [20]. Recently, scRNAseq-based analysis of TACs has revealed the existence of three TAC clusters in the mouse cornea, categorized by the differentiation stage based on their proliferative marker gene expression profile [70]. These are named early TACs (in the outer limbus), highly proliferative TACs (inner limbus) and mature TACs (cornea). Thus, it is possible that some of the cells from these varied TAC and/or LESC populations appear to be corneal epithelial progenitor cells [71].

Other studies have suggested that the progenitor-like cells present within the cornea are remnants of SCs from fetal/developmental stages. This idea originates from studies by Chang et al. (2011), wherein they showed that although central corneal epithelial cells do have stem-cell-like properties like LESCs, their stemness decreases with age [66]. Furthermore, studies in rats have shown that during development, ‘stem-like’ cells reside throughout the basal layer of the corneal epithelium; however, they become restricted to the limbus postnatally [72]. A similar post-natal loss of stem cells from the central cornea has been observed in mosaic mouse corneas, wherein the transition to the LESC-maintained corneal epithelium occurs between postnatal weeks 5 and 8 and the pattern is not fully mature until 10 weeks [73,74]. Tanifuji-Terai et al. (2006) have suggested that the mouse corneal epithelium may not be fully mature until 3–6 months after birth [75]. Thus, the identified corneal epithelial progenitor-like cells could be SCs remaining from early developmental stages. Furthermore, Mort et al. [23] suggested that in younger individuals, stem-like cells in the cornea could be SCs that persisted from fetal stages [75], while in older individuals, the progenitor-like cells could be the early TACs derived from LESCs [23].

## 5. Markers of Corneal TACs

One possible approach to establishing the identity of the progenitor-like cells within the cornea would be to use LESC/TAC-specific markers. Over the years, there has been a great effort by many groups to identify specific markers of LESCs and TACs. To date, many positive and negative markers have been identified for LESCs and TACs. However, the suitability and specificity of many of these markers are highly controversial, with many conflicting reports [76]. Importantly, there is currently no well-accepted set of markers that can distinguish between LESCs and early TACs. During homeostasis, LESCs have been proposed to possess certain defining features, such as a slow turnover rate, expression of certain proteins, expression of a specialized niche, clonogenicity, proliferative potential, and characteristic morphology [27], but many of these are characteristic of TACs too. Hence, specific molecular markers and characteristics that can unequivocally differentiate between LESCs and TACs are still needed. Some of the suggested biomarkers of LESCs and/or TACs (reviewed in [77]) are listed in Table 1.

Although, over past decade or so, a handful of markers were identified for LESCs and TACs [79,81,102,103], in recent years, with the advent of RNA sequencing (scRNAseq), various novel LESC and TACs markers have been proposed. Given that many reviews have nicely summarized the putative markers of TACs and LESCs based on earlier studies [79,81,102,103], herein, we have focused on the more recently proposed markers based on scRNAseq, summarized in Table 2.

Using scRNAseq, three TAC clusters in the mouse cornea were identified and categorized by the differentiation stage based on their proliferative marker gene expression profile [70]. An early TAC cluster, designated as TAC I, expressed high levels of Ki-67 (Mki67)^high^/baculoviral IAP repeat containing 5 (Birc5)^high^/CK15^mod^/activating transcription factor 3 (Atf3)^high^/metallothionein 1 (Mt1)^high^ expression, with one of the top differentially expressed gene (DEG) being the uracil DNA glycosylase (UNG) gene. A highly proliferative TAC cluster, designated as TAC II, identified primarily in the inner limbus, was shown to express a moderate level of CK15 with kinetochore-localized astrin/SPAG5-binding protein (Knstrn) as a top DEG. Finally, a mature TAC population, designated as TAC III, was identified in the cornea with high expression levels of PDZ-binding kinase (PBK). This corroborated earlier studies in humans that suggested PBK as a marker of TACs in the cornea, but not in the limbus. Furthermore, a unique cell cluster (containing 3.21% of the total of 16,360 limbal basal cells of human donor cornea) were identified as TACs in another study based on the expression of proliferation marker genes and a less differentiated progenitor status [104]. Out of the top 50 DEGs identified for TACs by scRNAseq, about 86% are cell cycle-dependent, suggesting the cell cycle-dependent genes may serve as signature markers of TACs [104]. Centromere protein F (CENPF), nucleolar and spindle-associated protein 1 (NUSAP1), ubiquitin-conjugating enzyme E2C (UBE2C), and cell division cycle 20 (CDC20) have all been identified as potential markers of human TACs, with UBE2C and CDC20 being suggested as the most specific markers since they represent the mitotic exit checkpoint genes [104]. Furthermore, the glycoprotein hormone subunit alpha 2 (GPHA2) was found to regulate the undifferentiated state of a population of cells identified as human limbal progenitor cells (LPCs) and thus could be used as a human LESC marker [105]. In another scRNAseq-based study in mice, high expression of thioredoxin-interacting protein (Txnip) and PBK was identified in the stem cell/early TAC and mature TAC populations, respectively, and they were proposed as novel regulators of stem cell and early TA cell quiescence [106]. Another study identified MKI67, survivin (BIRC5) and H2A histone family member X (H2AFX)-positive cells with differential expression of CD109 as highly proliferative TACs in human corneas, which had exclusive expression of cyclin-dependent kinase 2 (CKS2), stathmin-1 (STMN1), and UBE2C, which can be taken as their markers [107]. Finally, scRNAseq analysis of human iPSC-derived corneal organoids at 1, 2, 3, and 4 months of development identified that early TACs express CENPF, UBE2C, and NUSAP1 [108]. Many of these proposed markers that are based on scRNAseq analysis still need to be validated in different species and using combinatorial approaches in order for the research community to reach a consensus on reliable, exclusive, and universal markers of TACs and LESCs. Di and colleagues recently summarized the potential markers of human LESCs that were recently identified by scRNAseq as tumor protein 63 (TP63) and C-C motif chemokine ligand 20 (CCL20) for limbal stem/progenitor cells (LSPCs) with high stemness; GPHA2 and keratin 6B (KRT6B) for LSPCs with high differentiation; tetraspanin-7 (TSPAN7), SRY-box transcription factor 17 (SOX17), selectin E (SELE), endothelial cell surface-expressed chemotaxis and apoptosis regulator (ECSCR), receptor activity modifying protein 3 (RAMP3), ribonuclease A family member 1 (RNASE1), Niemann-Pick disease type C1 (NPCD1), nicotinamide N-methyltransferase (NNMT), solute carrier family 2 member 3 (SLC2A3), Krüppel-like factor 2 (KLF2), pyruvate dehydrogenase kinase 4 (PDK4) for LESCs [109] (37342216). Similarly, in mice, *Gpha2*, *Cd63*, interferon-induced transmembrane protein 3 (*Ifitm3*) for qLSCs and *Atf3*, suppressor of cytokine signaling 3 (*Socs3*), *Mt1*, PR domain zinc finger protein 1 (*Prdm1*) for aLSCs have been proposed [109].

**Table 2 cells-13-00748-t002:** Proposed markers of LESCs and corneal TACs based on scRNAseq.

Cell Type	Marker	Location/Species	Reference(s)
Early TACs (TAC I)	Ki-67 (Mki67)^high^/baculoviral IAP repeat-containing 5 (Birc5)^high^/CK15^mod^/activating transcription factor 3 (Atf3)^high^/metallothionein 1 (Mt1)^high^ expression with uracil DNA glycosylase (UNG) as top differentially expressed gene (DEG)	Human (outer limbus)	[70]
Highly proliferative TACs (TAC II)	Moderate K15 expression with kinetochore-localized astrin/SPAG5-binding protein (Knstrn) as a top DEG	Human (inner limbus)	
Mature TACs (TAC III)	High expression levels of PDZ-binding kinase (PBK)	Human (cornea)	
TACs	Centromere protein F (CENPF), nucleolar and spindle-associated protein 1 (NUSAP1), ubiquitin-conjugating enzyme E2C (UBE2C), and cell division cycle 20 (CDC20)	Human	[104]
Early TACs	Thioredoxin-interacting protein (Txnip)	Mouse	[106]
Mature TACs	PDZ binding kinase (PBK)	Mouse	
Highly proliferative TACs	Cyclin-dependent kinase 2 (CKS2), stathmin-1 (STMN1), and UBE2C	Human	[107]
Early TACs	CENPF, UBE2C, and NUSAP1	Human (iPSC-derived corneal organoids)	[108]
Limbal stem/progenitor cells (LSPCs) with high stemness	tumor protein 63 (TP63) and C–C motif chemokine ligand 20 (CCL20)	Human	[109]
LSPCs with high differentiation	GPHA2 and keratin 6B (KRT6B)	Human	
LESCs	Tetraspanin-7 (TSPAN7), SRY-box transcription factor 17 (SOX17), selectin E (SELE), endothelial cell surface-expressed chemotaxis and apoptosis regulator (ECSCR), receptor activity modifying protein 3 (RAMP3), ribonuclease A family member 1 (RNASE1), Niemann-Pick disease type C1 (NPCD1), nicotinamide N-methyltransferase (NNMT), solute carrier family 2 member 3 (SLC2A3), Krüppel-like factor 2 (KLF2), pyruvate dehydrogenase kinase 4 (PDK4)	Human	
qLSCs	*Gpha2*, *Cd63,* interferon-induced transmembrane protein 3 (*Ifitm3*)	Mouse	
aLSCs	*Atf3*, suppressor of cytokine signaling 3 (*Socs3*), *Mt1*, PR domain zinc finger protein 1 (*Prdm1*)	Mouse	

## 6. TACs in Other Organs: A Secondary Source of Information

The mechanisms by which adult SCs and TACs maintain homeostasis and regeneration have been studied in a number of other tissues, such as the skin, gastro-intestinal tract (including teeth, liver, pancreas, and salivary glands), respiratory tract, skeletal and cardiac muscles, nervous system, pituitary gland, kidney, breast, prostate, endometrium, mesenchyme, and bone marrow [110]. Herein, we discuss how TACs function to maintain tissue homeostasis and enable regeneration of other tissues, with the goal of understanding the potential roles and mechanisms of regulation of TACs in the cornea and identifying whether reversible transition/dedifferentiation of TACs and SCs has been observed in other tissues/organs.

### 6.1. Hair Follicles

Hair follicles (HFs) are considered a complex mini-organ and thus can serve as an ideal model for studying SCs and TACs [111]. HFs contain a diverse pool of SCs, such as epithelial stem cells, mesenchymal stem cells and melanocyte stem cells, which are located in different anatomical compartments, namely, the infundibulum, isthmus and lower follicle (bulge, germ and dermal papilla), respectively. To maintain healthy hair, the HF undergoes cycles in three stages, the anagen stage that involves the downward growth of the HF as it develops a new hair, the catagen stage that involves regression of the HF to a mature HF, and the telogen stage that is a resting stage. During telogen, two distinct populations of HF-SCs exist: the bulge SCs within the bulge region and hair germ SCs. The bulge SCs are more quiescent and cycle infrequently (quiescent SCs), while the germ SCs are sensitive to activation (primed SCs). During telogen, both SC populations are quiescent; however, as the HF transitions from telogen to anagen, the germ SCs are activated to proliferate, forming TACs that further differentiate to produce a new pool of matrix progenitor cells and downstream components, including the hair shaft, companion layer and inner root sheath [112]. The bulge SCs are also activated, giving rise to a downward-growing outer root sheath. The fate of the TACs is regulated by the dermal papilla. At the end of anagen, hair matrix proliferation ceases and most of the cells undergo apoptosis as the HF cycles into catagen. During catagen, the outer root sheath of the lower HF undergoes apoptosis and the keratinocytes of the upper outer root sheath collapses around the hair club (terminal structure of hair) and forms the bulge region and the secondary germ [113].

The bulge SCs were the first population of HF-SCs discovered in the 1990s [114] and have been shown to be Krt15+ and to give rise to all the lower epithelial cell lineages of the HF, i.e., outer root sheath cells, matrix cells, the companion layer, three layers of inner root sheath cells, the hair cuticle, the cortex and medulla. In the bulge, melanocyte stem cells derived from the neural crest also reside, supplying melanocytes to the hair matrix during each hair cycle. The melanocyte SCs and progenitor stem cells express two of the melanin-synthesizing enzymes, dopachrome tautomerase (Dct) and oxygenase tyrosinase-related protein 1 (Trp1), but they lack tyrosinase (Tyr), whereas in the anagen phase, the progenitors differentiate into mature melanocytes that express all three required enzymes (Trp1, Dct and Tyr). Differentiated melanocytes then migrate to anagen hair bulbs and make pigments in newly formed hair [115]. Hair germ SCs differ from bulge SCs in their proliferation potential and do not express Krt15 and CD34 but do express high levels of P-cadherin. The mesenchymal components of the HF, i.e., the dermal sheath, also contain SCs called dermal sheath mesenchymal stem cells or HF dermal stem cells that are capable of self-renewal and regeneration of the dermal sheath compartment after catagen [116]. The HF-associated sebaceous glands also contain Lrig1+ SCs and SCD1+ proliferative progenitor cells that differentiate into sebocytes [117]. The markers of different SCs within the HF components include CD200 (human), CD34, K19, Sox9, Lgr5, Hopx, Nfatc, Tcf, Lhx2 and Gli1 for the bulge SCs; Lgr6, Lrig1 and MTS24 for the isthmus SCs; Blimp1 for the sebaceous gland and Sca1 for the infundibulum [118]. Various factors and signaling pathways involved in the regulation of HF-SCs and TACs are listed in Table 3. Interestingly, the HF-TACs have been shown to perform functions other than their proliferative roles too. For example, HF-TACs, while generating their own progeny, also orchestrate a neighboring lineage for dermal adipogenesis through secreting sonic hedgehog (SHH) [119]. Also, they have been shown to have efferocytic roles in addition to a proliferative role [120]. It is worth exploring if all the TACs in other tissues can perform such moonlighting activities as seen with HF-TACs.

Another fascinating feature of HF-TACs is that dedifferentiation is integral to homeostatic stem cell maintenance. A recent study showed that melanocyte stem cells toggle between the SC and TAC states and can reversibly enter distinct differentiation states depending on local microenvironmental cues, e.g., WNT [121]. Similar dedifferentiation also takes place from early progenitor germ cells to bulge stem cells in the HF during both homeostasis and post-injury [122]. Is such a dedifferentiation integral to all TAC populations across organs or species? If not, what cues regulate them? It is also worth mentioning here that in the quiescent stem cells of HFs, micro-niches do exist, with spatio-temporal blueprints that can prime the TACs for differentiation into specific lineages. These micro-niches guide the complex lineages, which expand with the growth of the tissue [123]. The corneal SC and TAC research needs further investigation for identification and characterization of possible micro-niches.

**Table 3 cells-13-00748-t003:** Factors and signaling pathways involved in the regulation of HF-SCs and TACs.

Factor/Pathway	Species	Role	Reference
β-catenin	Human	Differentiates HF-SCs to HF-TACs by activating c-myc and regulating the expression of HF-TAC markers K15, K19, a6-integrin and β1-integrin	[124]
BMP signaling and pSMAD1/5 targets, e.g., Gata3	Mouse	Promotes HF-TAC lineage progression	[125]
miR-214/EZH2/β-catenin	Human	Regulates HF-SC proliferation and differentiation	[126]
Serum/glucocorticoid-regulated kinase family member 3 (Sgk3)	Mouse	Reduces the supply of HF-TACs, causing premature entry into the apoptotic regression phase of the hair cycle	[127]
T-cell leukemia/lymphoma protein 1 (Tcl1)	Mouse	Affects the cycling and self-renewal of HF-SCs and HF-TACs	[128]
Stable β-catenin-induced Wnt signaling pathway activation	Mouse	Causes transient activation of lymphoid enhancer-binding factor 1 (Lef1)/Tcf complexes that promote TAC conversion and proliferation	[129]
Transient activation of c-Myc	Mouse	Shifts keratinocytes from the SC to TAC compartment and thus stimulates proliferation and differentiation	[130]
β1 integrin signaling	Human	Maintains the survival, proliferation, apoptosis, and migration of human epithelial progenitors	[131]
Prostaglandin E2	Mouse	Attenuates the apoptosis of HF-TACs by promoting G1 arrest	[132]
Sonic hedgehog pathway	Mouse	Reinstalls dermal papilla for HF neogenesis	[133]
Notch/RBP-J Signaling	Mouse	Regulates the cell fate determination of hair follicular stem cells at the bulge region	[134]

### 6.2. The Testis in Mammalian Models

The mammalian testis serves as a powerful and elegant model to study stem cells due to: (i) its structural organization, which makes it possible to trace the progeny of individual stem cells, and (ii) the ease of analyzing the reconstitution of the stem cell population after an insult by studying the regeneration of this population or through transplantation experiments [135]. Both spermatogonia stem cells (SSCs) and TAC progenitors have been identified in the mouse testis [136], where the main function of the progenitor cells is to produce a large number of differentiated daughter cells, which are required for the continuous daily production of millions of motile sperm. Based on the pulse-chase of the undifferentiated spermatogonia, Nakagawa et al. (2007) [136] demonstrated that in mice, the spermatogenic stem cell system contains ‘actual stem cells’ and a second population called ‘potential stem cells’, which are essentially transit-amplifying cells. This distinction is based on the theory by Potten and Loeffler (1990) [137], which in principle says that irrespective of the cell types, the immediate progeny of the actual stem cells can retain ‘stemness’ while undergoing differentiation and can thus be referred to as ‘potential stem cells’. Importantly, not all TACs are potential stem cells. For differentiating spermatogonia, a major part of the transit-amplifying compartment is incapable of colony formation after transplantation [138,139]. In mice, a population of undifferentiated spermatogonia has been identified that express Miwi2 and behave as TACs during homeostasis; however, they retain stem cell-like properties and are critical for regenerative spermatogenesis [140].

### 6.3. Germline SCs, the Drosophila Model

The *Drosophila* is another excellent model for stem cell research, with a similar turnover rate to germline stem cells in males [141] and females [142,143]. TACs have been shown to dedifferentiate and become functional SCs under an appropriate microenvironment in the *Drosophila* testis [30]. The germline SCs lacking the Janus kinase-signal transducer and activator of transcription (Jak-STAT) pathway, which otherwise maintains the stem cells, differentiate into spermatogonia. Restoration of Jak-STAT signaling using conditional manipulation induces the dedifferentiation of TACs and spermatogonia to germline SCs. Furthermore, differentiated 4- or 8-cell interconnected germline cysts/cystocytes, generated either in the second instar larval ovaries of *Drosophila* or in adults over-producing the BMP4-like stem cell signal decapentaplegic (dpp), have been shown to efficiently convert into single stem-like cells [31]. These dedifferentiated cells can form functional germline stem cells and support normal fertility. The composition of the male germline stem cell system is well-conserved between mice and *Drosophila*, suggesting that germline stem cells in other organisms may also be capable of dedifferentiation [136]. Spermatogonial stem cells (SSCs) in rhesus macaques use a different strategy to meet a similar biological demand compared to rodents [144]. Unlike rodents’ testes, which have a small germline SC pool and a relatively larger pool of TA progenitors, rhesus testes have a larger SC pool with a relatively smaller TA progenitor compartment. Whether the other primate species, including humans, also have a larger SSC pool, as observed with the rhesus macaques, needs further investigation. The widely accepted factors/pathways involved in regulating SCs and TACs of mammalian testis and *Drosophila* germline are listed in Table 4.

### 6.4. Intestine

The intestinal epithelium is continuously exposed to harsh conditions that can lead to cell damage, such as digestive enzymes, non-physiological pH, and pathogens. Thus, the intestinal epithelium is endowed with various properties that allow regeneration and maintenance of homoeostasis, including self-renewing capabilities that are conferred by populations of SCs that are in an anatomically protected location within the crypts and villi. Two models of an intestinal stem cell (ISC) have been proposed: (i) the ‘stem cell zone model’ by Cheng and Leblond, which suggests the crypt base columnar (CBC) cells to be resident stem cells (active ISCs), and (ii) the ‘+4 model’ by Potten, which proposes the cells immediately above the Paneth cells to be stem cells (reserve/quiescent ISCs) [174]. CBCs express Leucine Rich Repeat-Containing G Protein-Coupled Receptor 5 (Lgr5), which is generally accepted as a specific marker of CBCs, besides others, such as CD44, Musashi-1 (Msi-1), Olfactomedin-4 (Olfm4), Achaetescute-like 2 (ASCL2), SPARC-related modular calcium-binding protein-2 (SMOC2), SOX9 and Krüppel-like factor 5 (KLF5) [175]. The rapidly cycling Lgr5^+^ ISCs give rise to TACs at the crypt–villus junction, which further differentiate into goblet, Paneth, enterocytes and enteroendocrine cells. The rapid cycling and differentiation of TACs in different zones of the intestinal crypts are regulated by adenomatous polyposis coli (APC), Ca^2+^ and calcium-sensing receptor (CaSR)-driven differentiation in the middle and upper crypt and apoptosis on the mucosal surface [176].

In the case of injury-induced ablation of Lgr5^+^ cells, the SCs located at the +4 position (i.e., reserve/quiescent ISCs) compensate for their function. These +4 position ISCs express Bmi, besides other markers, such as leucine-rich repeats and immunoglobulin-like domains 1 (Lrig1), mouse telomerase reverse transcriptase (mTert), homeodomain-only protein X (HOPX), inhibitor of differentiation 1 (ID1) and doublecortin-like kinase 1 (DCLK1) [175]. Thus, two ISC subpopulations imply that *Lgr5*^+^ CBCs are active SCs that mediate homeostatic self-renewal, whereas *Bmi1*+ quiescent SCs represent both a reserve SC pool in case of injury and a source for the replenishment of the Lgr5-expressing cells [177,178]. Besides this so called ‘reserve stem cell’ model of regenerative Bmi1+ quiescent SCs, a ‘dedifferentiation model’ has also been widely accepted, which says that plastic non-ISCs present in the TA zone can also dedifferentiate to active ISCs during injury-induced regeneration [179]. Murata et al. (2020) have shown that nearly all regeneration after ISC injury occurs by ASCL2-dependent dedifferentiation of recent LGR5+ cell progeny [180].

Besides dedifferentiation, another interesting feature of intestinal TACs is their vital role in the tuning of the differentiated cell-type composition. Using enteroid monolayers, 3D organoids and in vivo murine models, Sanman et al. (2021) have shown the existence of anticorrelation between progenitor cell proliferation and the ratio of secretory to absorptive cells. They found fewer rounds of cell division for secretory than absorptive progenitors, which suggests a ‘differential amplification model’ by which modulation of TAC proliferation, for example, during injury, infection, or calorie restriction, can control the tissue-cell-type composition. Such an underappreciated differential amplification role of TACs in other organs/species, such as the human cornea, is yet to be explored [181].

Intestinal TACs express markers that are shared with ISCs, such as prominin-1 (Prom1)/CD133 [182], polycomb transcriptional repressor, Bmi1 [183], inhibitor of DNA binding 1 (ID1) [184] and specific markers such as PR domain containing 16 (PRDM16) [185], transcription factor CCAAT/enhancer-binding protein α, C/EBPα [186], E3 ubiquitin ligase F-box and WD repeat domain-containing 7 (Fbw7) [187], protein arginine methyltransferase, PRMT1 [188]. The factors and pathways involved in regulating the transition between SCs and TACs in the intestine are summarized in Table 5.

### 6.5. Tooth

Many studies have revealed the interaction between SCs and TACs during homeostasis and regeneration, but they are all exclusively ectodermal organs [219]. The interaction between SCs of mesenchymal origin (MSCs) and their TACs has not been studied in as much detail when compared to epithelial stem cells [219]. Mesenchymal stem cells were first identified in teeth in 2000 and termed post-natal dental pulp stem cells (DPSCs) [220]. Subsequently, more types were identified, i.e., stem cells from the exfoliated deciduous (SHED) [221], periodontal ligament stem cells (PDLSCs) [222] and stem cells from the apical papilla (SCAP) [223] and dental follicle precursor cells (DFPCs) [224]. MSCs have an essential role in tooth development, homeostasis, and regeneration. Unlike adult stem cells of epithelial origin that are mostly unipotent, MSCs are multipotent and contribute toward the formation of various dental tissues, including dentin, pulp and periodontal structures. Huang et al. have elaborated and compared the dental MSCs with those from other tissues/organs [225]. As with adult stem cells of epithelial origin, MSCs undergo asymmetric cell divisions to produce another MSC that remains in the MSC pool and a TAC (called MTACSs) with increased proliferative capabilities that will differentiate into the different tooth cell types. The IGF-WNT signaling cascade is involved in MSC to TAC differentiation, whereas a Wnt5a/Ror2-mediated non-canonical WNT signaling pathway has been shown to be involved in the TAC to MSC feedback [226]. Using a mouse incisor as a model, Walker et al. have shown that distinct MSCs contribute to incisor MTACs. The MTAC feedback regulates the homeostasis and activation of cord-lining MSCs (CL-MSCs) through the Delta-like 1 homolog (Dlk1). This way, a balance between the MSC-MTAC number and the lineage differentiation is regulated [219].

Importantly, two different MSCs from human dental tissues, i.e., DFPCs and DPSCs, have been shown to revert/dedifferentiate to a naïve stem-cell-like status after osteogenic differentiation. Interestingly, dedifferentiated DSCs showed an enhanced potential to further differentiate toward the osteogenic phenotype compared to their undifferentiated counterparts [227]. The factors and pathways involved in regulating the transition between SCs and TACs in the tooth are summarized in Table 6.

### 6.6. Olfactory Epithelium

The olfactory epithelium (OE) is a specialized neuroepithelium lining the postero-dorsal aspect of the nasal cavity that possesses high regenerative capacity, unlike other parts of the nervous system. This high regenerative capability of the OE indicates the existence of an SC population [243]. In 1979, using 3H-thymidine-based autoradiography of the rat OE, Graziadei and Monti-Graziadei (1979) reported that the basal cells proper of the OE are SCs and the globose basal cells are possible transitional forms between the basal cells proper and the olfactory cells [243]. Now, it is well accepted that two sub-population of basal cells exist in the OE, the horizontal/dark basal cells (HBCs) and the globose/light basal cells (GBCs). In a normal, uninjured OE or even after ablation of the olfactory bulb, HBCs remain mitotically inactive/quiescent, indicating that they do not function as neuronal progenitors. The GBCs, on the other hand, progress from stem cell to differentiating olfactory sensory neuron (OSN) in four cell stages: (1) *Sox2* and *Pax6*-expressing stem cells (GBC_STEM_), (2) proneural transcription factor Ascl1(*Mash1*)-expressing early progenitor cells, GBC_TA-OSN_ (3) Neurogenin 1 (*Ngn1*)-expressing late-stage transit-amplifying cells, also called immediate neuronal precursors (GBC_INP_), and (4) postmitotic Neural cell adhesion molecule (*Ncam*)-expressing olfactory receptor neurons (ORNs) [243,244,245,246,247].

Lin et al. (2017) have shown that in the mouse OE, Ascl1+ progenitors and Neurog1+-specified neuronal precursors can dedifferentiate into multipotent stem/progenitor cells after epithelial injury, which can contribute significantly to tissue regeneration [248]. In another study, Gadye et al. (2017) have shown that following injury, the quiescent olfactory stem cells of mouse OE rapidly shift to activated, transient states, which are unique to injury-induced regeneration. One such fate is renewal of HBCs (or differentiation from a transient state), which can further differentiate [249]. Both the studies have also shown that Sox2 is required to initiate this dedifferentiation. The factors and pathways involved in regulating the transition between SCs and TACs in the OE are summarized in Table 7.

## 7. Conclusions

Taken together, substantial data have emerged to suggest that, under certain conditions, a subset of cells within tissues have the capability to dedifferentiate into stem-cell-like or progenitor-like cells to regenerate damaged stem cells pools. In tissues, as stem cells undergo asymmetric cell division, they produce TACs with high proliferative capabilities that will ultimately differentiate to produce mature cell types within the tissue. Substantial data have shown that in various tissues, a subset of the produced TACs retain progenitor-like properties, and that under certain conditions, they can dedifferentiate. Thus, a subset of TACs exist in a reversible state, and certain factors act as a pendulum dictating how the TACs transition between the stem cell and differentiated cell states.

## Figures and Tables

**Figure 1 cells-13-00748-f001:**
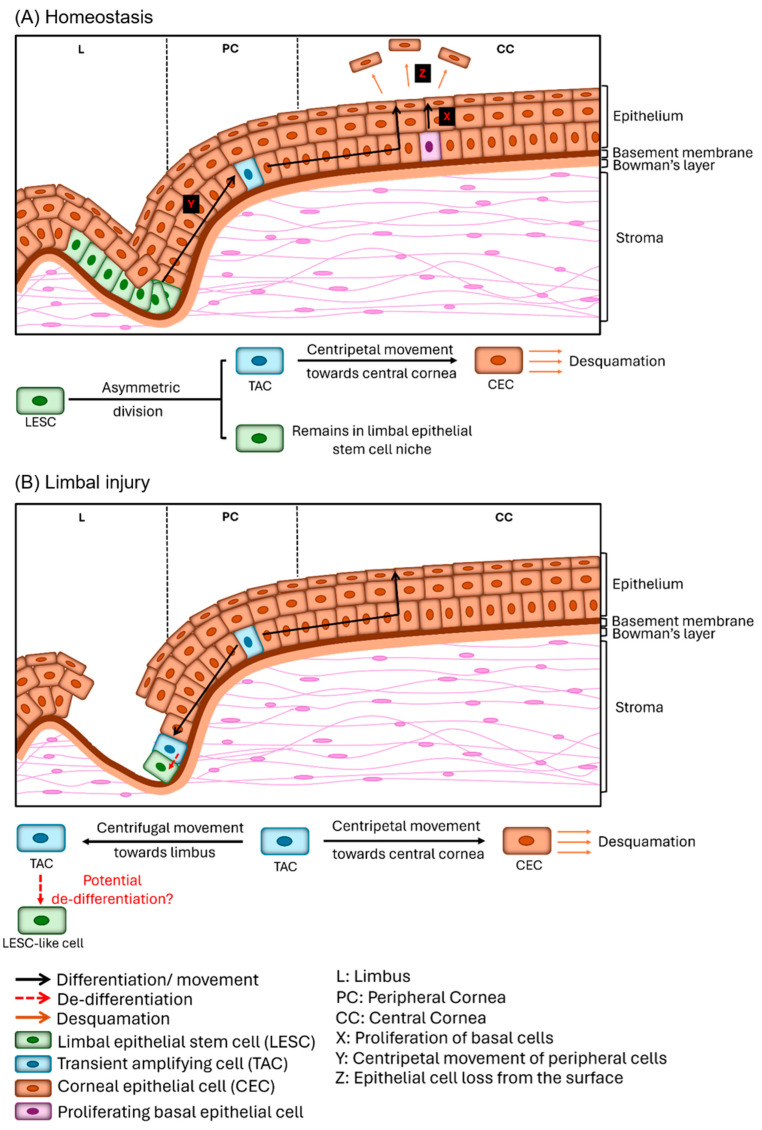
Corneal epithelial regeneration during homeostasis (**A**) and limbal injury (**B**). During homeostasis (**A**), a limbal epithelial stem cell (LESC) undergoes asymmetric division to produce a transient-amplifying cell (TAC) and another LESC. The TACs move centripetally through the peripheral cornea toward the central cornea to replenish the corneal epithelial cells that slough off, while the LESC remains within the limbal stem cell niche (LSCN). According to the XYZ hypothesis [25], corneal homeostasis is maintained by balancing the epithelial cell loss by cells moving in from the limbal region and by the proliferation of basal epithelial cells. Thus, if the proliferation of basal epithelial cells is X, the contribution to the cell mass of the centripetal movement of peripheral cells is Y, and the epithelial cell loss from the surface is Z, then corneal epithelial maintenance can be defined as: X + Y = Z. However, in the case of limbal injury, leading to loss of LESCs (**B**), studies have shown that TACs and corneal epithelial cells can move centrifugally toward limbus to resurface the limbal epithelium. Some groups have speculated that a subset of these cells that move centrifugally from the cornea into the limbus, possibly the TACs, can potentially dedifferentiate into LESC-like cells.

**Table 1 cells-13-00748-t001:** Proposed putative markers of LESCs and TACs.

Cell Type	Putative Marker	Species	Reference(s)
LESC (negative markers)	Cytokeratin 3 (CK3)	Rabbit	[26]
	Cytokeratin 12 (CK12)	Human	[78,79]
	Connexin 43	Human	[78,79]
	Involucrin	Human	[78]
	E-cadherin	Human	[78]
	NGF receptor (p75NTR)	Human	[78]
	Nestin	Human	[79]
LESC (positive markers)	N-terminal truncated form of p63α (ΔNp63α)	Human	[80]
	CCAAT-enhancer-binding protein δ (C/EBP δ)	Human	[28]
	Polycomb complex protein-Bmi1	Human	[81]
	ATP-binding cassette sub-family B member 5 (ABCB5)	Human, mouse	[82]
	ATP-binding cassette super-family G member 2 (ABCG2)	Human, rabbit, rat	[32,47,78,79,83,84]
	Nerve growth factor (NGF) and its receptors tropomyosin receptor kinase A (TrkA)	Human	[85]
	GDNF family receptor alpha-1 (GFRα-1)	Human	[86]
	Musashi-1	Human	[87]
	Notch-1	Human	[88]
	Integrin α9	Human, mouse	[78,89,90]
	Cytokeratin 15 (CK15)	Human, mouse	[91]
	Cytokeratin 14 (CK14)	Mouse	[92]
	Cytokeratin 19 (CK19)	Human, mouse	[91,92]
	Wnt family member 4 (Wnt-4)	Human	[93]
	SRY-box transcription Factor 9 (Sox9)	Human	[94,95]
	Alpha-actinin-1 (Actn1)	Mouse	[94]
	Frizzled class receptor 7 (Fzd7)	Human, mouse	[94,96]
	Cytokeratin 17 (CK17)	Mouse	[94]
	N- and P-cadherin	Human	[97]
	Vimentin	Human	[79]
TAC markers	α9β1 integrins	Mouse	[98,99]
	Basal cell adhesion molecule (BCAM)	Human, mouse	[100]
	α-enolase	Human, rabbit	[99,101]
	Connexin-43	Human	[78,79,99]
	Cytokeratin 19 (CK19)	Human	[99]

**Table 4 cells-13-00748-t004:** Factors/pathways involved in regulating the SCs and TACs of mammalian testis and *Drosophila* germline.

Factor/Pathway	Species	Role	Reference(s)
Lola	Male *Drosophila*	Maintains SCs and germ cell differentiation	[145]
Profilin		Maintains SCs germ cell enclosure by somatic cyst cells	[146]
Notch and Delta		Required for survival of the germline stem cell lineage	[147]
Held-out-wings (HOW)		Maintains SC maintenance and controls the onset of transit-amplifying divisions	[148]
Dynein-light-chain-1 (DDLC1/LC8)		Regulates spermatogonial divisions	[149]
CG8005		Mediates TACs’ spermatogonial divisions via oxidative stress	[150]
Epidermal growth factor signaling		Regulates the differentiation of germline cells	[151]
CG6015		Controls the TACs divisions by EGFs signaling	[152]
E-cadherin-based adherens junctions		Regulates asymmetric stem cell division	[153]
Maelstrom (Mael)		Differentiates the GSC lineage	[154]
dBigH1 and bag of marbles (Bam)		Regulates SC differentiation	[155]
Terminal uridylyl transferase 1 (tut), bag of marbles (bam), or benign gonial cell neoplasm (bgcn)		Coordinates the proliferation and differentiation	[156]
ERK downstream targets		Regulates TACs and subsequent differentiation of neighboring germline cells	[157]
Bam		Switches from proliferation to terminal differentiation in TACs	[158]
Wnt/b-catenin signaling		Mediates proliferation of undifferentiated spermatogonia (SSCs and TACs)	[159]
Transforming growth factor-beta (TGFb)		Regulates SC maintenance and TAC proliferation	[160]
Insulin/IGF signaling, TOR signaling, and GCN2-dependent amino acid sensing		Promotes the proliferation and maintenance of the stem/progenitor population	[161,162]
Rac family small GTPase (Rac)	Female *Drosophila*	Mediates polarity to ensure a robust pattern of asymmetric division	[163]
Tumor suppressor brain tumor (Brat)		Regulates the linker histone dBigH1 expression	[164]
Niche-derived Hh and Wnts and germline-derived EGFs		Promotes the differentiation of GSCs	[165]
Insulin/IGF signaling, TOR signaling, and GCN2-dependent amino acid sensing		Promotes the proliferation and maintenance of the stem/progenitor population	[161,162]
Retinoic acid-STRA8 signaling	Mouse testis	Regulate spermatogenesis by controlling spermatogonial differentiation and meiotic initiation	[166]
Netrin-1 receptor UNC5C		Contributes to the homeostasis of undifferentiated spermatogonia	[167]
Tumor suppressor gene Rb		Required for self-renewal of spermatogonial stem cells	[168]
Glial cell line-derived neurotrophic factor		Regulates spermatogonial stem cells	[169]
Breast cancer-amplified sequence 2 (BCAS2)		Involved in alternative mRNA splicing in spermatogonia and the transition to meiosis	[170]
Mammalian target of rapamycin complex 1 (mTORC1)		Required for spermatogonial differentiation	[171]
SH2 domain-containing protein tyrosine phosphatase-2 (SHP2)		Required for the transition from stem cell to progenitor spermatogonia and male fertility	[172]
SOX3		SOX3 promotes the generation of committed spermatogonia in postnatal testes	[173]

**Table 5 cells-13-00748-t005:** Factors/pathways involved in regulating the SCs and TACs of the intestine.

Factor/Pathway	Species	Role	Reference(s)
Wnt/β-catenin-based suppression of the mitogen-activated protein kinase (MAPK)	Mouse	Balances proliferation and differentiation in ISCs	[189,190]
K-Ras	Mouse	Promotes expansion and hyperproliferation of TACs	[191]
BMP signaling	Mouse	Dampens Lgr5^+^ ISC renewal	[192]
Delta1-Notch signaling	Mouse	Controls the secretory commitment of TACs through lateral inhibition	[193]
Hippo signaling	Mouse	Deletion of Lats1/2 (Hippo kinases) results in the loss of Lgr5^+^ ISCs and expansion TACs	[194]
Growth factor signaling such as epidermal growth factor receptor (EGFR)/ErbB1	Human	Major drivers of proliferation in the ISC niche	[195]
Cytokeratin-8 (K8)-regulated Notch signaling	Mouse	Promotes differentiation of TACs	[196]
IL-10 (rmIL-10)	Mouse and ISC cultures	Expands the number of TACs and enhances the differentiation	[197]
Interleukin 22 via inhibition of Notch and Wnt signaling	Mouse and ISC cultures	Expands TACs	[198]
Methyltransferase 3, N6-Adenosine-Methyltransferase (METTL3)	Mouse	Survival of TACs	[199]
Lysophosphatidic acid receptor 5 (LPA5) receptor	Mouse	Survival of SCs and TACs	[200]
Mixed-lineage leukemia 1 (MLL1/KMT2A)	Mouse	Loss of MLL1 is accompanied by loss of ISCs and differentiation bias toward the secretory lineage	[201]
Krüppel-like factor 5 (Klf5)	Mouse	Maintains proliferation of both CBCs and TACs	[202,203]
Prefoldin RPB5 interactor (URI)	Mouse	Helps in survival and differentiation of TACs	[204,205]
Survivin	Mouse	Survival of TACs	[206]
Death receptor 5 (DR5)	Mouse	Survival of TACs	[207]
Cyclin/CDK inhibitor p57^Kip2^	Mouse	Maintains Hopx^+^ ISC quiescence	[208]
Foxl1^+^ mesenchymal cells	Mouse	Maintains proliferation of ISCs and TACs	[209]
Polycomb group (PcG) proteins	Human	Repress the terminal differentiation in the TACs	[210]
Myeloid translocation gene-related 1 (MTGR1)	Mouse	Maintains the ISCs in an undifferentiated state	[211]
CBL family ubiquitin ligases	Mouse	Maintain ISCs	[212]
Rho GTPase family member, CDC42	Mouse	CDC42 deletion leads to diminished ISCs and highly expanded TACs	[213]
Src42a and Src64b	*Drosophila*	Required for ISC divisions	[214]
Unfolded protein response (UPR)	Mouse	Required for SC to TAC transition	[215]
Lipopolysaccharide (LPS)	Mouse	Represses cell proliferation through RIPK3-mediated necroptosis of ISCs and TACs	[216]
Hypomorphic X-box–binding protein 1 (Xbp1)	Mouse	Increases ISC numbers	[217]
Intraepithelial lymphocytes (IELs)	Mouse	Modulate the proliferation of TACs	[218]

**Table 6 cells-13-00748-t006:** Factors/pathways involved in regulating the SCs and TACs of the tooth.

Factor/Pathway	Species	Role	Reference(s)
Arid1a	Mouse	Regulates the fate of TACs by limiting proliferation, promoting cell cycle exit and differentiation	[228]
Runt-related transcription factor (Runx)2+/glioma-associated oncogene (Gli)1+ cells via insulin-like growth factor IGF signaling	Mouse	Maintain MSC niche, regulates proliferation and differentiation of TACs and growth rate of the incisor tooth	[229]
MAPK and PI3K pathways	Mouse	Regulate dental epithelial stem cell activity, transit-amplifying cell proliferation, and enamel formation	[230]
Notch and FGF signaling	Mouse and organ culture model	Decide the fate of SCs in incisors	[231,232]
TGF-βI (Alk5)	Mouse	Regulates the proliferation of TACs and maintenance of SCs	[233]
Prominin-1 (Prom1/CD133)	Mouse	Absence results in the disruption of stem cell quiescence maintenance and activation	[234]
Polycomb repressive complex 1 (PRC1)	Mouse	Regulates the TAC phenotype by controlling the expression of key cell-cycle regulatory genes and Wnt/β-catenin signaling	[235,236]
E-cadherin	Mouse	Inactivation leads to decreased label-retaining stem cells, decreased cell migration and increased proliferation	[237]
c-Myb	Mouse	Involved in differentiation	[238]
Transforming growth factor-beta 1 (TGF-β1) and connective tissue growth factor (CTGF)	Mouse	Involved in the functioning of TACs during incisor development (embryonic day 16.5–post-natal day 3.5)	[239]
Yap	Mouse	Maintains proliferation and inhibits differentiation	[240]
YAP/TAZ and mTOR signaling	Mouse	Drive the proliferation of TACs	[241]
CXCR4/CXCL12 signaling	Mouse	Activates enamel progenitor cell division and controls the movement of epithelial progenitors from the dental stem cell niche	[242]

**Table 7 cells-13-00748-t007:** Factors/pathways involved in regulating the SCs and TACs of the OE.

Factor/Pathway	Species	Role	Reference
Testicular receptor 2, Nr2c1	Mouse	Involved in regulating the progenitor or early differentiation state	[250]
Fibroblast growth factors (FGF)	Mouse	Delay differentiation of a committed neuronal TAC (the INP) and support proliferation or survival of a rare cell, possibly a stem cell, that acts as a progenitor of INPs	[251]
Transforming growth factor beta (TGF-β)	Mouse	Plays key roles in feedback loops to regulate the size of progenitor cell pools, and thereby the neuron number, during development and regeneration	[252]
Bone morphogenetic protein (BMP4)	OE cultures from mouse	Inhibits proliferation of MASH1-expressing progenitors when present at high concentrations and stimulates survival of newly generated ORNs when present at low concentrations	[253]
De novo methyltransferase DNmt3b	Mouse	Plays a role in the initial steps of progenitor cell differentiation	[254]
Zinc finger transcription factor Insm1	Mouse	Promotes the transition of progenitor cells from proliferative apical to terminal, neurogenic basal cells	[255]

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
