# Peer review of "The Potential Reversible Transition between Stem Cells and Transient-Amplifying Cells: The Limbal Epithelial Stem Cell Perspective"

_cells, 2024, doi:10.3390/cells13090748_

Round 1

Reviewer 1 Report

Comments and Suggestions for Authors

This manuscript is an excellent review on the state of the art respective to limbal stem cells and its progeny, designed as Transient Amplifying Cells (TACs). The only point that could rise discussion is related with the possible "dedifferentiation" of the TACs to behave newly as Stem cells. Such issue is controversial since the differential gene expression that could be found between LESCs and early progenitors (early TACs) is so difficult to establish. Perhaps, those cells that could be found as "stem cells" at central cornea could correspond to early progenitors that maintain many of the markers and proliferative potential found or expected for real stem cells (see Sun et al., 2010, Nature 463:E10). On the other hand it is possible that the limbal niche does not possess an uniform distribution as suggested by previous results which demonstrate that limbus has a heterogeneous structure (Wiley et al, 1991, Invest Ophthalmol Vis Sci 32:594-602). Under authors criteria, it could be convenient (but not mandatory) to discuss more explicitly the influence of the basal membrane composition on the expression of stem cell characteristics, and on the establishment of the stem cell niche (Bains et al., 2023, Cells. 12:2334; Polisetti et al., 2016, Stem Cells. 34:203-219; Funderburgh et al., 2016, Ocul Surf. 14:113-120;  Yamada et al., 2015, Mol Vis. 21:1328-39). 

Author Response

We thank the reviewer for their time evaluating our manuscript and are sincerely thankful for their important comments and suggestions. We have added the recommended references, except Wiley et al, 1991, Invest Ophthalmol Vis Sci 32:594-602. We apologize for this, but this study is rather outdated and we found it hard to incorporate the findings into our review. Our changes have been tracked by highlighting the added text in yellow.

Reviewer 2 Report

Comments and Suggestions for Authors

The manuscript reviews the evidence regarding the role of transient amplifying cells (TACs) as potential progenitor cells as well as evidence regarding the role of differentiation in tissue regeneration in different organs, including the cornea. This is a comprehensive review that has been structured and written very well. The references are relevant and used appropriately. The topic is important and I believe that the article will be useful for the readers.

I have no comments regarding the quality of the manuscript. As a suggestion, a schematic figure showing the location of the LESCs and TACs and the XYZ movements could help understand the topic.

Author Response

We sincerely thank the reviewer for their time and valuable suggestions. We have added the suggested figure to the manuscript as Figure 1.